# Comprehensive Analysis of Gut Microbiota Composition and Functional Metabolism in Children with Autism Spectrum Disorder and Neurotypical Children: Implications for Sex-Based Differences and Metabolic Dysregulation

**DOI:** 10.3390/ijms25126701

**Published:** 2024-06-18

**Authors:** Amapola De Sales-Millán, Paulina Reyes-Ferreira, José Félix Aguirre-Garrido, Ismene Corral-Guillé, Rehotbevely Barrientos-Ríos, José Antonio Velázquez-Aragón

**Affiliations:** 1Doctorado en Ciencias Biológicas y de la Salud, Universidad Autónoma Metropolitana, Ciudad de México 09340, Mexico; 2212803115@correo.ler.uam.mx; 2Departamento de Salud Mental, Instituto Nacional de Pediatría, Ciudad de México 04530, Mexico; paulinareyesferreira@gmail.com; 3Departamento de Ciencias Ambientales, Universidad Autónoma Metropolitana-Lerma, Lerma 52006, Estado de Mexico, Mexico; j.aguirre@correo.ler.uam.mx; 4Centro de Investigación del Neurodesarrollo, Instituto Nacional de Pediatría, Ciudad de México 04530, Mexico; ismenepuma@yahoo.com.mx; 5Laboratorio de Citogenética, Instituto Nacional de Pediatría, Ciudad de México 04530, Mexico; rehotbevely@gmail.com; 6Laboratorio de Oncología Experimental, Instituto Nacional de Pediatría, Ciudad de México 04530, Mexico

**Keywords:** autism spectrum disorder, dysbiosis, gut microbiota, gut microbiota sex differences, metabolic dysregulation

## Abstract

This study aimed to investigate the gut microbiota composition in children with autism spectrum disorder (ASD) compared to neurotypical (NT) children, with a focus on identifying potential differences in gut bacteria between these groups. The microbiota was analyzed through the massive sequencing of region V3–V4 of the 16S RNA gene, utilizing DNA extracted from stool samples of participants. Our findings revealed no significant differences in the dominant bacterial phyla (Firmicutes, Bacteroidota, Actinobacteria, Proteobacteria, Verrucomicrobiota) between the ASD and NT groups. However, at the genus level, notable disparities were observed in the abundance of *Blautia*, *Prevotella*, *Clostridium XI*, and *Clostridium XVIII*, all of which have been previously associated with ASD. Furthermore, a sex-based analysis unveiled additional discrepancies in gut microbiota composition. Specifically, three genera (*Megamonas*, *Oscilibacter*, *Acidaminococcus*) exhibited variations between male and female groups in both ASD and NT cohorts. Particularly noteworthy was the exclusive presence of *Megamonas* in females with ASD. Analysis of predicted metabolic pathways suggested an enrichment of pathways related to amine and polyamine degradation, as well as amino acid degradation in the ASD group. Conversely, pathways implicated in carbohydrate biosynthesis, degradation, and fermentation were found to be underrepresented. Despite the limitations of our study, including a relatively small sample size (30 ASD and 31 NT children) and the utilization of predicted metabolic pathways derived from 16S RNA gene analysis rather than metagenome sequencing, our findings contribute to the growing body of evidence suggesting a potential association between gut microbiota composition and ASD. Future research endeavors should focus on validating these findings with larger sample sizes and exploring the functional significance of these microbial differences in ASD. Additionally, there is a critical need for further investigations to elucidate sex differences in gut microbiota composition and their potential implications for ASD pathology and treatment.

## 1. Introduction

Autism spectrum disorder (ASD), is a group of heterogeneous disorders, characterized by a delay or alteration in the acquisition of skills in a variety of developmental domains including motor, social, language and cognition [1], manifested before the age of three [2]. According to a recent report, ASD prevalence in 2018, revealed that ASD affects approximately 23 per 1000 children under 8 years in the United States and 22.5 per 1000 in Hispanic children [3]. In particular, ASD prevalence in Mexico was estimated to be 8.7 cases per 1000 [4]. ASD is more prevalent in men than women, and in the USA, it is diagnosed more frequently in boys than girls, the ratio varies depending on the state from 3.3 to 5.2:1 in 2018 [3], in England with data from 2016 the estimate male–female ratio was 3:1 [5] and most recent ratio obtained from USA data is 4.3:1 [6].

Gastrointestinal issues, in particular, constipation, diarrhea, and abdominal pain, are among the most common comorbidities diagnosed in individuals with ASD. Compared to children with neurotypical (NT) development, who exhibit neurological development and functioning consistent with age-appropriate norms, the overall prevalence of gastrointestinal (GI) symptoms in children with ASD is 55% [7]. The frequent GI symptoms in individuals with ASD suggest a potential connection between the gut microbiota and the clinical symptoms of the disorder (gut–brain axis), often correlating with the severity of ASD [8].

The heterogeneity of the symptoms and severity of ASD has made analysis complex of possible changes in children’s microbiota since the results in different studies are often controversial. At the phylum level, the main differences found between studies were changes in the abundance of Firmicutes, Bacteroidota, Proteobacteria, Actinobacteria, and Verrucomicrobia [7].

At the genus level, the genera that presented changes in their abundance were *Bifidobacteria*, *Lactobacillus*, *Coprococcus*, *Roseburia*, *Clostridium*, *Faecalibacterium*, *Oscillospira*, *Ruminococcus*, *Dialister*, *Veillonella*, *Prevotella*, *Escherichia*/*Shigella*, *Sutterella*, *Phyllobacterium*, *Flavonifractor*, *Nitriliruptor* and *Collinsella*. However, these changes at both the genus and phylum levels are not conclusive since some studies presented increases and many others decreased abundances [7,9]. Further research is necessary to strengthen the evidence and identify the specific genera and species altered in patients with ASD, considering populations with diverse cultural and genetic backgrounds. The aim of this study was to characterize the gut microbiota of children with ASD compared to neurotypical (NT) children, in order to identify potential alterations in gut microbiota composition.

## 2. Results

### 2.1. Participant Characteristics

The group of individuals with ASD consisted of 30 subjects, comprising 25 males and 5 females, with an average age of 6 years and 8 months. The NT group consisted of 31 individuals, consisting of 26 males and 5 females, with a mean age of 7 years and 8 months. In terms of the severity level of core symptoms of autism, 9 individuals (30%) were classified as mild severity, 13 (43%) were classified as moderate severity, and 8 (27%) were classified as severe. Among the five female participants, four exhibited mild symptoms. All participants, both in the ASD and NT groups, resided in Mexico City.

### 2.2. Gut Microbiota Composition in Children with ASD and NT

The analysis of the microbial composition at the phylum level in both ASD and NT groups revealed that Firmicutes (ASD: 60.9% vs. NT: 59.6%), Bacteroidota (ASD: 17.1% vs. NT: 15.1%), Actinobacteriota (ASD: 15.4% vs. NT: 17.2%), Proteobacteria (ASD: 4% vs. NT: 3.6%), and Verrucomicrobiota (ASD: 1.3% vs. NT: 2.7%) constituted the predominant phyla in the gut microbiota of both groups. However, no significant differences were observed in the microbial composition at this taxonomic level between the ASD and NT groups (Figure 1A).

At the genus level, *Bifidobacterium* was the most abundant genus in both the ASD and NT groups, accounting for 11.83% and 14.44%, respectively. Although a decrease was observed in patients with ASD, no significant difference was found by Welch’s *t*-test. Other relatively abundant genera included *Roseburia* (ASD: 11.64% vs. NT: 6.88%), *Faecalibacterium* (ASD: 8.70% vs. NT: 5.09%), *Bacteroides* (7.67% vs. 8.79%), and *Lachnospiraceae_unclassified* (ASD: 7.24% vs. NT: 5.13%), with no significant differences detected between the groups (Figure 1B).

Furthermore, only four taxa at the genus level exhibited significant differences in their relative abundances between the ASD and NT groups. These taxa were *Blautia* (ASD: 3.46% vs. NT: 9.80%, *p* < 0.02), *Prevotella* (ASD: 2.54% vs. NT: 0.71%, *p* < 0.03), *Clostridium_XI* (ASD: 0.67% vs. NT: 0.59%, *p* < 0.01), and *Clostridium_XVIII* (ASD: 0.39% vs. NT: 0.97%, *p* < 0.01) (Figure 1B–F).

### 2.3. Differences for Sex in the Gut Microbiota at All Levels between ASD vs. NT

Sex-based differences in gut microbiota composition between individuals with ASD and NT children were examined across various taxonomic levels, ranging from phylum to genus. The study cohort was stratified by sex, and comparative analyses were conducted among the four resultant groups. A total of 20 phyla, 40 classes, 44 orders, 85 families, and 128 genera were identified. Significance testing for differential microbiota abundance was performed using the Statistical Analysis of Metagenomic Profiles (STAMP) tool.

Across the taxonomic hierarchy, significant differences were observed in microbiota composition. Specifically, at the class level, two classes, *Negativicutes* (*p* < 0.02) and *Erysipelotrichia* (*p* < 0.05), exhibited significant disparities between the groups. Similarly, at the order level, two orders, *Selenomonadales* (*p* < 0.02) and *Erysipelotrichiales* (*p* < 0.05), demonstrated significant differences. At the family level, *Erysipelotrichaceae* (*p* < 0.05) exhibited discrepant abundances. Furthermore, at the genus level, three genera, namely *Megamonas* (*p* < 0.01), *Oscilibacter* (*p* < 0.04), and *Acidaminococcus* (*p* < 0.01), displayed significant variations among the groups (see Figure 2A–E). Of particular note is the exclusive presence of the genus *Megamonas* in the ASD female group, with a relative abundance of 4.45%. Notably, this genus was absent in ASD males as well as both male and female NT groups.

### 2.4. Differences in Gut Microbiota among Patients with ASD Based on Severity Levels

An analysis of the gut microbiota composition in patients with ASD based on their severity levels (mild, moderate, and severe) at the phylum level revealed the following distributions: Firmicutes (ASD-Mild: 61.19% vs. ASD-Moderate: 63.39% vs. ASD-Severe: 56.66%), Actinobacteria (ASD-Mild: 20.49% vs. ASD-Moderate: 9.45% vs. ASD-Severe: 19.50%), Bacteroidota (ASD-Mild: 13.60% vs. ASD-Moderate: 18.47% vs. ASD-Severe: 18.96%), Proteobacteria (ASD-Mild: 1.69% vs. ASD-Moderate: 6.03% vs. ASD-Severe: 3.20%), and Verrucomicrobiota (ASD-Mild: 1.25% vs. ASD-Moderate: 1.72% vs. ASD-Severe: 0.87%). These phyla constituted the predominant components of the gut microbiota across the three groups. However, no significant differences in the gut microbial composition were observed at this taxonomic level based on the severity scale (Figure 3A).

At the genus level, the same analysis revealed that *Bifidobacterium* and *Faecalibacterium* were the most abundant genera in the three ASD groups, accounting for 16.18%, 7.10%, and 17.68%, and 10.46%, 9.65%, and 4.56%, respectively. Although a decrease in *Bifidobacterium* was observed in patients with moderate ASD and a decrease in *Faecalibacterium* in those with severe ASD, no significant differences were found. Other relatively abundant genera included *Bacteroides* (ASD-Mild: 7.79% vs. ASD-Moderate: 10.19% vs. ASD-Severe: 7.15%), *Lachnospiraceae incertae sedis* (ASD-Mild: 4.81% vs. ASD-Moderate: 2.98% vs. ASD-Severe: 4.87%), *Roseburia* (ASD-Mild: 4.68% vs. ASD-Moderate: 15.35% vs. ASD-Severe: 9.35%), *Blautia* (ASD-Mild: 4.55% vs. ASD-Moderate: 2.79% vs. ASD-Severe: 3.43%), *Ruminococcaceae unclassified* (ASD-Mild: 4.12% vs. ASD-Moderate: 2.60% vs. ASD-Severe: 3.70%), *Phascolarctobacterium* (ASD-Mild: 3.89% vs. ASD-Moderate: 3.20% vs. ASD-Severe: 2.33%), and *Clostridiales unclassified* (ASD-Mild: 3.77% vs. ASD-Moderate: 2.95% vs. ASD-Severe: 3.50%), with no significant differences detected among the three groups. Furthermore, only one taxon at the genus level exhibited significant differences in relative abundance among the three ASD groups: *Lactobacillus* (ASD-Mild: 0.05% vs. ASD-Moderate: 0% vs. ASD-Severe: 0.09%, *p* < 0.01) (Figure 3B).

### 2.5. Community Diversity in the ASD and NT Groups

For alpha diversity metrics, (Chao, Shannon, and Simpson indexes, (Figure 4A–C) the ASD group showed greater richness than the NT group. Only the Chao index showed significant differences (*p* < 0.01, Figure 3A). Regarding beta diversity, no statistically significant differences were discerned across taxonomic levels ranging from phylum to genus (Figure 5A–E).

### 2.6. Metabolic Functional Inferred Analysis of Microbiome in ASD and NT Groups

A total of 376 microbial MetaCyc pathways were employed to construct the functional profile. Following the exclusion of pathways with low occurrence rates (positivity rates < 10%), 32 pathways exhibited significant differences between the ASD and NT groups. Among these, thirteen pathways were found to be enriched in the ASD group compared to the NT group, as depicted in Figure 6.

Within the subset of thirteen enriched pathways in the ASD group, four were associated with the biosynthesis or degradation of amine and polyamine (ARG+POLYAMINE-SYN, POLYAMINESYN3-PWY, POLYAMSYN-PWY, PWY-6478, and PWY0-1241). Additionally, two pathways participated in fermentation processes leading to the production of short-chain fatty acids (PWY-5677 and PWY-7013), while one pathway pertained to aromatic compound degradation (P281-PWY). Furthermore, one pathway was identified for vitamin biosynthesis (PWY-5920), one for energy precursor production (PWY-7254), and one for amino acid degradation (THREOCAT-PWY).

Conversely, the pathways underrepresented in the ASD group compared to the NT group encompassed three related to amino acid biosynthesis and degradation (ARGORNPROST-PWY, PWY-2941, and PWY-5028), two involved in the biosynthesis and degradation of carbohydrates (FUCCAT-PWY and PWY-7315), and two related to carbohydrate fermentation (P164-PWY and PWY-7003). Additionally, six pathways were associated with nucleoside and nucleotide biosynthesis and degradation (PWY-6126, PWY-6220, PWY-6222, PWY-6229, PWY-5695, and PWY-6608), while one each was allocated for methanogenesis (METH-ACETATE-PWY), inositol degradation (PWY-7237), oxidation (P221-PWY), inorganic nutrient metabolism (PWY490-3), and teichoic acid biosynthesis (TEICHOICACID-PWY).

## 3. Discussion

The altered microbiome–gut–brain axis has been implicated in the etiology of neurodevelopmental disorders and has been proposed to exacerbate the core symptoms of ASD [8]. In a multi-level analysis, the microbiome–gut–brain axis exhibited microbial and molecular profiles associated with ASD [10]. Furthermore, an intervention involving *Lactobacillus reuteri* (6475 + 17,938) improved social behavior in patients with ASD [11]. These findings suggest a significant role of the microbiome–gut–brain axis in modulating social behavior. This modulation arises from the microbiome’s regulation of intestinal permeability, immune function, and metabolism of neurotransmitter precursors or other relevant metabolites for central nervous system function [12]. The gut microbiome has also been implicated in various mental disorders. It has been reported that gut microbes influence the immune response, and a chronic inflammatory state may increase susceptibility to stress, potentially leading to the development of stress-related disorders such as major depressive disorder [13]. Moreover, distinct microbial species are associated with schizophrenia, and altered functional microbiome profiles—particularly those related to short-chain fatty acids (SCFAs), neurotransmitters, and tryptophan metabolism—have been observed in patients with schizophrenia [14].

In recent years, a growing body of research has focused on the role of the microbiota and dysbiosis in the pathophysiology of ASD [15,16]. Multiple studies have reported differences in gut microbiota composition between ASD patients and NT children [7,10,17]. These differences manifest at various taxonomic levels, ranging from phylum [18] to genus [19]. Additionally, some studies have examined the levels of different metabolites, such as short-chain fatty acids (SCFAs), phenylalanine, and tyrosine metabolism, finding correlations between alterations in gut microbiota and metabolite levels [20,21,22].

Most studies comparing gut microbiota composition in ASD patients and neurotypical children involve relatively small sample sizes, typically around 30 participants per group [7,17,18,23]. However, some studies with larger sample sizes have identified significant differences, such as a study conducted in Korea involving 54 ASD patients and 38 neurotypical children found a significant decrease in Bacteroidetes abundance and an increase in Actinobacteria in the ASD group at the phylum level. At the genus level, the study reported a lower abundance of *Bacteroides* and *Bifidobacterium* in the ASD group. [24]. Another study in Australia with 99 ASD patients suggested an effect of ASD-related behaviors and dietary preferences on the microbiota composition of children with ASD, contrary to the claimed causal role of microbiome in ASD [25]. A recent meta-analysis including 690 ASD patients identified confounding variables but still revealed microbial signatures associated with ASD symptomatology [26]. Furthermore, a robust Mendelian randomization study confirmed a potential causal relationship between specific gut microbe changes and ASD [15].

In our study, we present the results of microbiota analysis in 30 prepubertal ASD children and 31 neurotypical children, matched for age mean age of seven years and eight months) and sex. This matching is crucial, as both age and sex are confounders in microbiota analyses [26]. In both groups, five girls were included, which is in accordance with the 4.5:1 affected males/affected females ratio reported for ASD [4,6]. We found no significant changes at the phylum level, Firmicutes and Bacteroidota being the most abundant phyla in both groups. However, significant differences were observed at the genus level, particularly for *Blautia*, *Prevotella*, *Clostridium XI*, and *Clostidium XVIII*. These genera have been consistently implicated in ASD-related gut microbiota alterations [23].

Several investigations comparing the microbiota of individuals with ASD and NT children have noted alterations in the Firmicutes/Bacteroidota ratio at the phylum level [7,10,19]. However, in our study, we observed a change in the Firmicutes/Bacteroidota ratio but we did not find a significant change at the phylum level. Firmicutes and Bacteroidota phyla exhibited a slight increase in ASD patients compared to NT children, and they were the most abundant phyla in both groups. Regarding alpha diversity in the ASD group, all indices were higher than NT, but only the Chao index displayed a statistical difference, while no differences were observed for the Shannon and Simpson indices. Similar results were found in other studies, where alpha diversity was greater in ASD patients versus NT patients [27]. Furthermore, beta diversity analysis revealed no disparities at any taxonomic level examined.

At the taxonomic genus level, significant differences between groups were evident for *Blautia*, *Prevotella*, *Clostridium XI*, and *Clostridium XVIII*. These genera have been consistently implicated in differences between ASD and NT children. *Clostridium* genus, in particular, has been extensively studied, with disparities reported since the seminal work of Finegold [28] up to recent investigations across various populations [29,30,31,32,33,34,35,36,37]. A meta-analysis conducted in 2020 highlighted significant differences in *Clostridium* abundance between ASD patients and NT children [23]. *Clostridium* is regarded as a potentially harmful bacterium, with its spore-forming ability leading to the release of pro-inflammatory toxins that can affect the central nervous system [38]. Additionally, a high abundance of *Clostridium* has been associated with lower levels of benzaldehyde, and a lower level in the gut of this compound may reflect an alteration in gut oxidative stress [39].

*Blautia*, a genus frequently reported to exhibit decreased abundance in children with ASD compared to NT children, plays a significant role in gut microbiota dynamics [29,30,34,35,36,37,40]. *Blautia* species are known for their capacity to produce butyrate, a short-chain fatty acid crucial for maintaining intestinal health and exerting anti-inflammatory effects [41]. Additionally, certain members of the *Blautia* genus possess the ability to metabolize dietary fiber from various plants into Polymethoxyflavones (PMFs), flavonoids known for their antioxidant, anti-inflammatory, and anticancer properties [42]. Notably, a specific strain of *Blautia*, *Blautia stercoris MRx0006*, demonstrated significant effects in ameliorating social deficits, repetitive behaviors, and anxiety-like behaviors in a mouse model of ASD, further highlighting the potential relevance of the *Blautia* genus in ASD pathophysiology [43].

On the other hand, *Prevotella* genus abundance has shown inconsistency in studies examining gut microbiota in children with ASD. Some studies reported increased *Prevotella* abundance [44], while others observed decreased abundance [30]. Interestingly, microbiota transfer therapy in children with ASD led to improvements in autism-related symptoms, coinciding with increased *Prevotella* abundance [36,45]. In our cohort of ASD patients, we also observed an increase in *Prevotella* abundance. The *Prevotella* genus exhibits genetic and metabolic diversity within and between species, potentially impacting the microbiota–gut–brain axis by promoting chronic inflammation, as evidenced in various models [41,46].

ASD disproportionately affects males compared to females; a phenomenon reflected in the limited number of females with ASD included in our study. Despite this small sample size, differences in microbiota structure at various taxonomic levels were evident. Specifically, at the genus level, three genera—*Megamonas*, *Oscilibacter*, and *Acidaminococcus*—exhibited significant variations among groups. Notably, the genus *Megamonas* was exclusively observed in our ASD female group. However, *Megamonas* has previously been observed in the guts of non-autistic individuals in other populations [47,48]. The absence of *Megamonas* in the other groups in our study may be attributed to factors such as the small sample size. *Megamonas* species ferment glucose into acetic acid and propionic acid, serving as substrates for lipogenesis and cholesterol synthesis, thereby providing energy for the host [48]. Furthermore, *Megamonas* has been associated with enrichment in obese adults and obese children with non-alcoholic fatty liver disease, linked to acetate production and triglyceride accumulation [47].

Sex constitutes a crucial confounder in microbiota analysis, exerting profound effects on the structure and physiology of the microbiome, gut, and brain at systemic and local levels [49]. A meta-analysis of gut microbiome studies in ASD patients revealed innate differences in bacterial taxa abundances associated with sex, underscoring the need to address sex bias in research due to the higher prevalence of ASD in males [26]. Numerous studies in both human and animal models have identified sex differences in gut microbial structure, largely attributed to the interplay between gut microbiota and steroids, which modulate neural activity. Gut microbiota also regulates the secretion of incretins and neurotransmitters and modulates vagus nerve–enteric nervous system communication [49].

Sex-associated differences in gut microbiota structure have been documented in Mexican children from rural and urban backgrounds. While the main disparities in gut microbiota were observed at the phylum level and associated with the children’s origin (rural vs. urban), differences were also noted between males and females regardless of their origin [50]. However, many studies lack a sex-differentiated analysis of microbiota, potentially introducing bias into the results obtained [49]. Therefore, future studies focusing on ASD girls and their microbiota are imperative to obtain a more precise understanding of the taxa that may be specifically altered in this population.

A comparison of the gut microbiota of patients with ASD according to their severity levels identified only one statistically significant difference at the genus level. The genus *Lactobacillus* was more abundant in patients with severe ASD and was practically absent in those with mild ASD. *Lactobacillus* is a common inhabitant of the gut, particularly in children, and is a group of lactic acid producers considered as probiotics. Interventions with some members of this genus have been shown to improve ASD symptoms. For example, *L. reuteri* has been shown to improve social behavior in children with ASD [11], and supplementation has produced improvements in the severity of ASD in both Italian [51] and Taiwanese populations [52].

Moreover, certain species of the *Lactobacillus* genus are reported to induce early proinflammatory cytokines such as IL-8, TNF-α, IL-12p70, and IL-6 [53]. Additionally, lactobacillus abundance in macaques has been associated with the inhibition of the enzyme Indoleamine 2,3-dioxygenase (IDO1), which ultimately affects Th17 cell functioning related to mucosal immune disruption [54]. A study in India reported a 32-fold increase in the abundance of the genus Lactobacillus in children with ASD compared to healthy children [55].

Furthermore, a high abundance of the *Lactobacillaceae* and *Bifidobacteriaceae* families has been associated with a decrease in SCFA-producing bacteria [56]. SCFAs are gut microbial-derived bacterial metabolites that play a critical role in the immune system, brain functioning, and behavior [8]. Our functional analysis observed that the metabolism of SCFAs is altered in our group of patients with ASD.

All this information suggests that while the *Lactobacillus* genus includes species that can ameliorate some ASD symptoms, it also contains species that can affect SCFA metabolism and promote a proinflammatory state.

In the functional analysis of the microbiome, we identified statistical differences in 32 metabolic pathways between individuals with ASD and NT individuals. It is noteworthy that this analysis relied on the inference of metabolic pathways based on the genera identified through 16S metagenomic analysis, rather than a more robust shotgun functional analysis. Nonetheless, we observed intriguing alterations enriched in ASD patients, with five pathways related to the degradation of amines and polyamines and one pathway associated with amino acid degradation. Amines serve numerous physiological roles, acting as neurotransmitters, local hormones, and regulators of various bodily functions, including gastric acid secretion, cell growth and differentiation, circadian rhythm regulation, and immune response [57,58,59,60]. The enrichment of these pathways in ASD patients is significant, given that amines are involved in the synthesis of key neurotransmitters such as dopamine, epinephrine, histamine, and serotonin, and imbalances in amines have been reported in ASD patients [61]. Furthermore, two pathways involved in fermentative SCFA production were found to be enriched in ASD patients. This enrichment aligns with previous reports of elevated SCFA levels, particularly propionic acid, in children with ASD, as recently reviewed by Lagod and Naser [62].

Conversely, we identified 19 metabolic pathways that were underrepresented in the microbiota of patients compared to NT children. Among these pathways, three were related to amino acid synthesis and degradation, echoing the implications discussed earlier regarding neurotransmitter metabolism. Additionally, four pathways were involved in the biosynthesis, degradation, and fermentation of carbohydrates. Impaired carbohydrate digestion and gut dysbiosis have been previously identified in children with ASD [63]. SCFAs are produced through the fermentation of fiber and carbohydrates in the intestines, along with long-chain fatty acids and tryptophan metabolites, and play a role in preventing inflammation [64]. The interplay between gut microbiota, inflammation, and SCFAs in ASD has been extensively reviewed elsewhere [8].

Our study was conducted with a limited sample size of patients with ASD and NT children, which may have limited the statistical robustness needed to detect a greater number of genera altered in the microbiota or microbiota metabolic pathways in children with ASD. Nonetheless, we identified relevant changes in the microbiota structure of ASD patients, involving genera previously reported to be altered in ASD gut microbiota and exhibiting metabolic consequences consistent with previous reports of altered metabolites or metabolic pathways in affected children. The identification of microbiota structural differences in female groups affected by ASD and NT children underscores the importance of sex-directed analysis of gut microbiota in ASD-affected children for future studies.

## 4. Materials and Methods

### 4.1. Participants of the Study

The research adopted a clinical–prospective–comparative design, with the primary objective of delineating the intestinal microbiota profiles of individuals diagnosed with ASD in comparison to NT counterparts. The participant cohort was exclusively drawn from the Mental Health service at the National Institute of Pediatrics, located in Mexico City. Stringent inclusion criteria were applied uniformly across both cohorts, ensuring homogeneity and comparability.

Enrolment criteria dictated that participants be children aged between 6 to 8 years, assessed with the Wechsler Intelligence Scale for Children-IV (WISC-IV), and attaining an intelligence quotient (I.Q.) equal to or exceeding 85. Diagnosis of non-syndromic ASD, categorized as level 1 or 2 according to the Diagnostic and Statistical Manual of Mental Disorders, Fifth Edition (DSM-5), was confirmed utilizing the Criteria Diagnostic Interview (CRIDI-ASD/DSM-5).

Neurotypical children, meticulously matched for age and gender with ASD-diagnosed counterparts, were recruited to form the control group. Selection criteria necessitated each NT child to correspond precisely in age and gender to an already included ASD-diagnosed participant. Recruitment of neurotypical participants was facilitated through invitations extended to siblings or acquaintances of patients receiving services from various departments within the institute. Subsequently, neurotypical status was ascertained following evaluation by the Neurodevelopmental Service of our institute, thus affirming their classification.

Recruitment activities for both ASD and NT groups transpired over a duration spanning from March 2021 to June 2023. Conversely, exclusion criteria were applied rigorously to maintain the homogeneity and integrity of the participant cohorts. Individuals presenting with any additional comorbidities, particularly syndromic disorders such as Fragile X Syndrome or Angelman Syndrome, were excluded from the study. Furthermore, participants exhibiting eating or food ingestion disorders outlined in the DSM-5, including pica, rumination, binge eating disorder, avoidance disorder, or intake restriction food, were ineligible for participation. Moreover, individuals who had consumed dietary supplements containing probiotics or prebiotics within the preceding six months, as well as those administered oral antibiotics during the same timeframe, were excluded. Such exclusions were imperative to mitigate confounding variables potentially influencing the composition of the intestinal microbiota.

### 4.2. Ethics

The parents or guardians of all participants accepted freely read and signed their informed consent in accordance with the Declaration of Helsinki revised in 2013 and Mexican regulations for clinical studies and biological waste handling and disposal. The ethics, biosafety, and scientific committee of the National Institute of Pediatrics approved the research protocol (SOL2020/64).

### 4.3. Stool Sample Collection, DNA Extraction and Sequencing

Fresh stool specimens, ranging from 1 to 5 g, were collected into sterile containers under the guidance of participants’ relatives, who received comprehensive instructions regarding the proper execution of this procedure. Subsequent to collection, the stool samples were promptly refrigerated at 4 °C to preserve their integrity during transport. Within a timeframe not exceeding 4 h, the samples were delivered to the laboratory, where aliquots of approximately 200 milligrams each were individually extracted and subsequently frozen at −70 °C until further processing.

Upon commencement of the DNA extraction phase, stool samples weighing 0.6 g were thawed to facilitate subsequent procedures. Metagenomic DNA extraction was carried out for a total of 61 stool specimens, adhering strictly to the protocols outlined in the QIAamp^®^ Fast DNA Stool Mini Kit (cat. 51604-50, QIAGEN^®^, Redwood, CA, USA). The quantification of purified stool DNA was performed by assessing its absorbance at 260 nanometers utilizing a NanoDrop Lite Spectrophotometer (Thermo Scientific, Waltham, MA, USA), while the quality of the extracted DNA was evaluated through electrophoretic fractionation employing 1.0% agarose gels.

For Illumina sequencing targeting the V3-V4 region of the 16S rRNA gene, DNA libraries were constructed utilizing 10 nanograms of DNA extracted from the stool samples. This process was executed utilizing the IIlumina MiSeq high-throughput sequencing platform (MiSeq, 300 + 300 base pairs, paired-end sequencing) at the Integrated Microbiome Resource situated within Dalhousie University, Canada. This rigorous methodology ensured the acquisition of high-fidelity sequencing data necessary for comprehensive microbiome analysis.

### 4.4. Bioinformatic Analysis

Microbiota analysis of stool samples was conducted using MOTHUR v.148.0 software, employing the Illumina MiSeq Standard Operating Procedure (SOP). To assess alpha diversity, rarefaction curves were generated by plotting the number of sequences (*x*-axis) against the number of operational taxonomic units (OTUs) (*y*-axis), ensuring that sequencing depth adequately captured the breadth of diversity present in the samples. Alpha diversity metrics, including richness, Chao, Shannon, and Simpson indices, were calculated and reported.

For beta-diversity analysis, STAMP v.2.1.3 software was utilized. ANOVA was applied to assess the relative abundance of sequences, while correlation coefficient matrices were computed for principal component analysis (PCoA). This comprehensive approach facilitated the exploration of inter-sample diversity patterns.

PICRUSt2 analysis was employed to predict metagenome function from amplicon sequencing data. Subsequently, the results of this analysis were utilized in conjunction with STAMP v.2.1.3 software to predict the composition of metabolic pathways, providing insights into the functional potential of the microbial communities.

To ensure transparency and facilitate reproducibility, all 16S rDNA sequence files and corresponding mapping files for the samples utilized in this study have been deposited in the NCBI BioSample repository. Interested parties may access these resources using the provided Accession Number PRJNA1103839, which can be found at the following link: https://www.ncbi.nlm.nih.gov/sra/PRJNA1103839.

### 4.5. Statical Analysis

For the statistical analysis of alpha diversity, SPSS Statistics v.22.0 was employed. Normality tests and homogeneity tests were conducted to assess the distribution and equality of variances, respectively, between groups. A significance threshold of *p* ≥ 0.05 was utilized to ascertain normality and homogeneity of variances. Subsequently, for normally distributed and homogeneous data, the Student’s *t*-test was applied, while for non-normally distributed or non-homogeneous data, the Mann–Whitney test was utilized. Statistical significance was defined as *p* < 0.05.

To analyze differences in the microbiome across phylum to genus levels and pathways, STAMP v.2.1.3 software was employed. Welch’s *t*-test was utilized to assess differences between groups, with multiple test corrections conducted using the Benjamini–Hochberg false discovery rate (FDR). Genera exhibiting low occurrence rates and expression levels (positivity rates < 10%) were excluded from the analysis to ensure robustness and reliability of the findings.

## 5. Conclusions

This study delineated discernible differences in genus-level abundance, specifically regarding *Blautia*, *Prevotella*, and *Clostridium*, within the microbiota of children diagnosed with ASD and NT children of Mexican origin. Notably, these genera have been previously implicated as perturbed in the microbiota of children with ASD, indicative of a proinflammatory milieu and altered SCFA’s metabolism. Such alterations are of significance in the context of the gut–brain axis functionality.

Moreover, our investigation unveiled sex-based distinctions in microbiota composition, even though based on a limited sample size comprising five girls with ASD and five NT girls. Particularly, the presence of the genus *Megamonas*, associated with cholesterol metabolism, was exclusive to the ASD female group.

Functional analysis further delineated group discrepancies in SCFA’s metabolism, amine and polyamine metabolic pathways, and carbohydrate and fiber fermentation. These findings collectively underscore the intricate interplay between microbiota composition and functional pathways, warranting further investigation to unravel the precise mechanistic underpinnings of microbiota–host interactions in ASD.

## Figures and Tables

**Figure 1 ijms-25-06701-f001:**
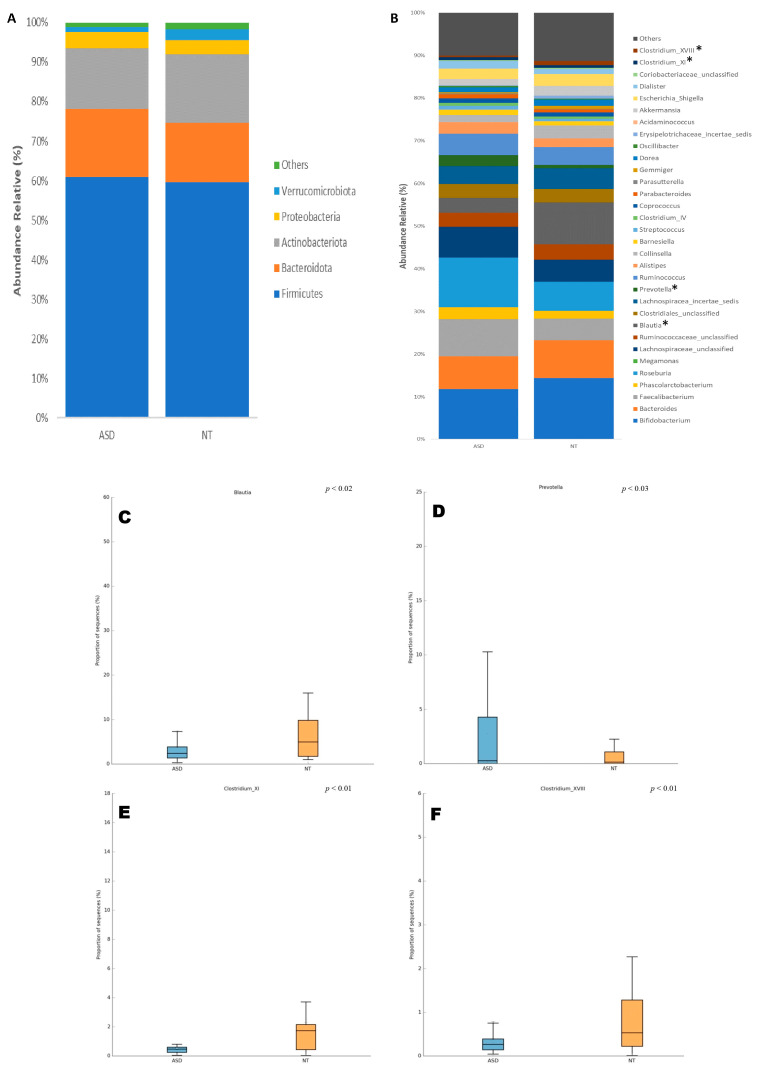
Composition of the gut microbiota at (**A**) phylum and (**B**) genus taxonomic levels in patients with ASD and NT children. Statistical differences were observed only at genus level for (**C**) *Blautia*, (**D**) *Prevotella*, (**E**) *Costridium XI*, and (**F**) *Clostridium XVII*. * *p* < 0.05 Welch’s *t*-test.

**Figure 2 ijms-25-06701-f002:**
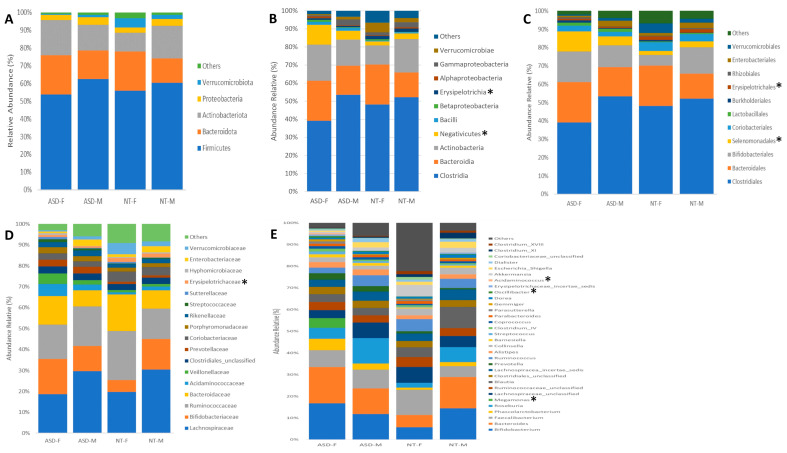
Composition of the gut microbiota at (**A**) phylum, (**B**) class, (**C**) order, (**D**) family, and (**E**) genus levels in patients with ASD and NT children divided by sex. Statistical differences were observed at genus level for (**F**) *Megamonas*, (**G**) *Oscilibacter,* and (**H**) *Acidaminococcus*. * *p* < 0.05 Welch’s *t*-test.

**Figure 3 ijms-25-06701-f003:**
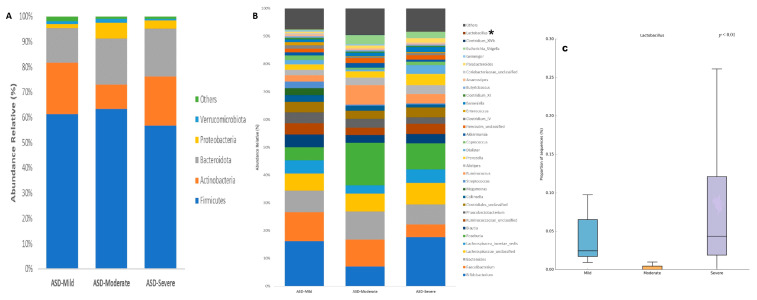
Composition of the gut microbiota of patients with ASD classified by level of severity at (**A**) phylum and (**B**) genus taxonomic levels. Statistical differences were observed at genus level only for (**C**) *Lactobacillus*. * *p* < 0.05 Welch’s *t*-test.

**Figure 4 ijms-25-06701-f004:**
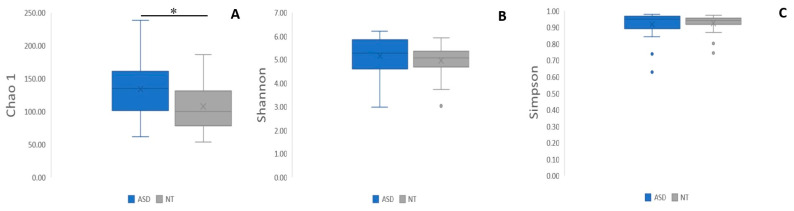
Differences in alpha diversity indices (**A**) Chao, (**B**) Shannon, and (**C**) Simpson. The box plot figures show the alpha diversity of the bacterial communities in the two study groups: ASD and NT control. * *p* < 0.01.

**Figure 5 ijms-25-06701-f005:**
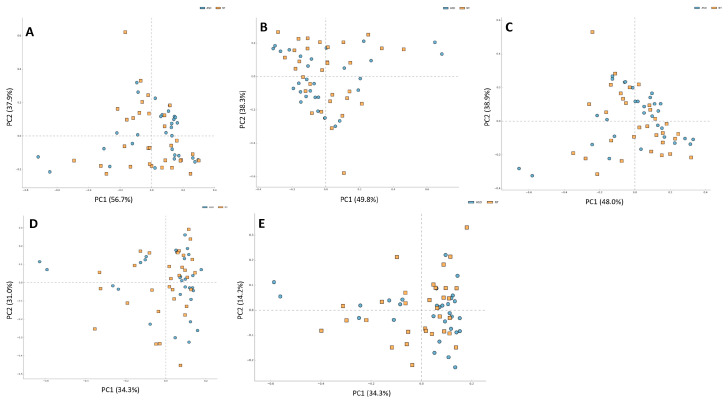
Differences in beta diversity at all levels (**A**) phylum, (**B**) class, (**C**) order, (**D**) family, and (**E**) genus in two groups of study: ASD and NT control.

**Figure 6 ijms-25-06701-f006:**
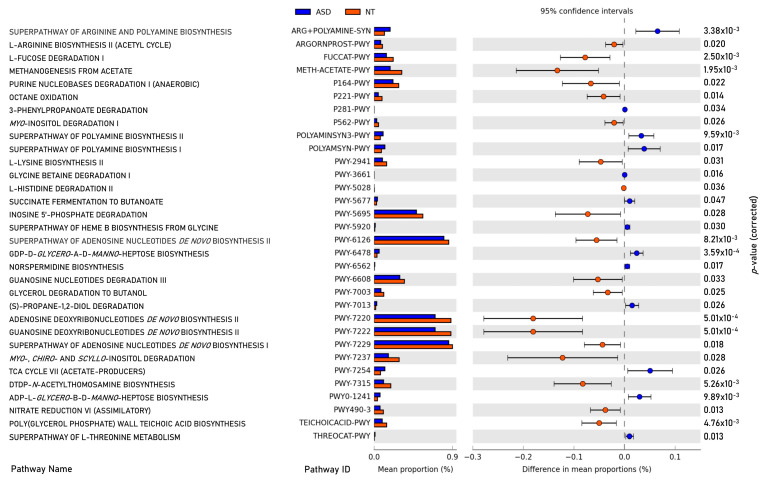
Significant differences in abundance in metabolic pathways of gut microbiota between children with ASD and NT controls (*p*-value corrected < 0.05).

## Data Availability

All 16S rDNA sequence files and corresponding mapping files for the samples utilized in this study have been deposited in the NCBI BioSample repository. Interested parties may access these resources using the provided Accession Number PRJNA1103839, which can be found at the following link: https://www.ncbi.nlm.nih.gov/sra/PRJNA1103839.

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
