# Peer review of "Comprehensive Analysis of Gut Microbiota Composition and Functional Metabolism in Children with Autism Spectrum Disorder and Neurotypical Children: Implications for Sex-Based Differences and Metabolic Dysregulation"

_ijms, 2024, doi:10.3390/ijms25126701_

Round 1
Reviewer 1 Report
Comments and Suggestions for Authors
Reviewer comments and suggestions
The authors in this study investigated the gut microbiota composition in children with autism spectrum disorder (ASD) compared to neurotypical (NT) children. The microbiota was analyzed through the massive sequencing of region V3-V4 of the 16S RNA gene, utilizing DNA extracted from stool samples of participants. Our findings revealed no significant differences in the dominant bacterial phyla (Firmicutes, Bacteroidota, Actinobacteria, Proteobacteria, Verrucomicrobiota) between the ASD and NT groups. However, at the genus level, notable disparities were observed in the abundance of Blautia, Prevotella, Clostridium XI, and Clostridium XVIII, all of which have been previously associated with ASD. Additionally, a sex-based analysis unveiled additional discrepancies in gut microbiota composition. Specifically, three genera (Megamonas, Oscilibacter, Acidaminococcus) exhibited variations between male and female groups in both ASD and NT cohorts. Particularly noteworthy was the exclusive presence of Megamonasin females with ASD. Analysis of predicted metabolic pathways suggested an enrichment of pathways related to amine and polyamine degradation, as well as amino acid degradation in the ASD group. This study's findings contributed to the growing body of evidence suggesting a potential association between gut microbiota composition and ASD.
Overall, the manuscript was good. However, a few major concerns/comments needed to be explained or modified.
- Line 56-57 is there was any reason for this. Please check the sentence
- Line 61 please explain the term neurotypical.
- Line 76-77 what does the line indicate please write briefly consisting of references if possible.
- Line 109 There were differences (phylum and genus) between these two, what would be the possible reason or if they are bit similar, did the authors discuss it comprehensively?
- Line 135-136 what does it indicate?
- Line 195-196 no need to mention here better to include the novelty they obtained in the first two to three lines.
- Line 200 please explain a bit based on the cited references.
- Line 212 reference number 28 please explain their results too.
- Line 250 is this word was correct DGOS.
- The conclusion should be short and suggestive for the future to be explored based on the present findings.
- Please check the references number 3, 16,18, 25, 56
Author Response
Answer to Reviewer 1
- Line 56-57 is there was any reason for this. Please check the sentence.
Answer: The sentence was delated, and the previous sentence was modified and incorporated to previous paragraph. The change was highlighted.
- Line 61 please explain the term neurotypical.
Answer: We establish as neurotypical children those who neurological development and functioning are consistent with what is typically expected within the general population. The paragraph was modified and highlighted.
- Line 76-77 what does the line indicate please write briefly consisting of references if possible.
Answer: The following modifications have been made to the paragraph:
"At the genus level, the genera that exhibited changes in their abundance were Bifidobacteria, Lactobacillus, Coprococcus, Roseburia, Clostridium, Faecalibacterium, Oscillospira, Ruminococcus, Dialister, Veillonella, Prevotella, Escherichia/Shigella, Sutterella, Phyllobacterium, Flavonifractor, Nitriliruptor, and Collinsella. However, these changes at both the genus and phylum levels are not conclusive, as some studies reported increases while many others reported decreases in abundance [10,11]. Further research is necessary to strengthen the evidence and identify the specific genera and species altered in patients with ASD, considering populations with diverse cultural and genetic backgrounds. The aim of this study was to characterize the gut microbiota of children with ASD compared to neurotypical (NT) children, in order to identify potential alterations in gut microbiota composition."
The changes were highlighted.
- Line 109 There were differences (phylum and genus) between these two, what would be the possible reason or if they are bit similar, did the authors discuss it comprehensively?
Answer: There is an error in the figure caption. The statistical differences were observed only at the genus level, which led to the confusion. Since the differences were only at the genus level, there was no discussion regarding differences at the phylum level.
The figure caption has been corrected and highlighted to prevent this confusion.
- Line 135-136 what does it indicate?
Answer: There is also an error in the figure caption. The original caption is imprecise and unclear. The intention of this figure is to illustrate the differences at various taxonomic levels between ASD patients and NT children, and to highlight the differences in gut microbiota composition between males and females, primarily at the genus level.
The figure caption has been corrected and highlighted to avoid this confusion.
- Line 195-196 no need to mention here better to include the novelty they obtained in the first two to three lines.
Answer: The sentence was deleted and replaced with the following:
"The altered microbiome-gut-brain axis has been implicated in the etiology of neurodevelopmental disorders and has been proposed to exacerbate the core symptoms of ASD [6]. In a multi-level analysis, the microbiome-gut-brain axis exhibited microbial and molecular profiles associated with ASD [9]. Furthermore, an intervention involving Lactobacillus reuteri (6475 + 17938) improved social behavior in patients with ASD [10]. These findings suggest a significant role of the microbiome-gut-brain axis in modulating social behavior."
The sentences were highlighted.
- Line 200 please explain a bit based on the cited references.
Answer: The sentence was deleted and replaced with the following:
"The gut microbiome has also been implicated in various mental disorders. It has been reported that gut microbes influence the immune response, and a chronic inflammatory state may increase susceptibility to stress, potentially leading to the development of stress-related disorders such as major depressive disorder [12]. Moreover, distinct microbial species are associated with schizophrenia, and altered functional microbiome profiles—particularly those related to short-chain fatty acids (SCFAs), neurotransmitters, and tryptophan metabolism—have been observed in patients with schizophrenia [13]."
The sentences were highlighted.
- Line 212 reference number 28 please explain their results too.
Answer: We have changed the sentence to:
"However, some studies with larger sample sizes have identified significant differences. For example, a study conducted in Korea involving 54 ASD patients and 38 neurotypical children found a significant decrease in Bacteroidetes abundance and an increase in Actinobacteria in the ASD group at the phylum level. At the genus level, the study reported a lower abundance of Bacteroides and Bifidobacterium in the ASD group [24]."
The sentences were highlighted.
- Line 250 is this word was correct DGOS.
Answer: The sentence was imprecise; we have changed it to:
"Additionally, a high abundance of Clostridium has been associated with lower levels of benzaldehyde. A lower level of this compound in the gut may reflect an alteration in gut oxidative stress [39]."
The sentence was highlighted.
- The conclusion should be short and suggestive for the future to be explored based on the present findings.
Answer: We shorted the conclusion; the paragraphs were highlighted.
- Please check the references number 3, 16,18, 25, 56
Answer: Reference 3 was revised and corrected to reflect a prevalence value of 23 ASD cases per 1,000 children under 8 years. The change was highlighted.
The text referring to reference 16 was eliminated along with the reference itself. The change was highlighted.
Reference 18, which is a review of the role of dysbiosis in ASD patients but does not provide original evidence, was deleted from the text and references. The change was highlighted.
Reference 25 reports an increase in p-cresol in the urine of ASD patients compared to controls. P-cresol is a metabolite of the phenylalanine pathway, and it is appropriate to cite this article in the paragraph.
Reference 56 is a review of the metabolic role of biogenic amines and aligns with the sentence in which it was used.
Note: Please consider that many references were eliminated, and therefore, the numbering of the references has changed.

Reviewer 2 Report
Comments and Suggestions for Authors
De Sales-Millán et al. compares the gut microbiota of neurotypical (NT) vs autism spectrum disorder (ASD) individuals by performing 16S sequencing of stool samples. They further analyze microbial differences between male and female participants. Given the very low numbers of female participants, it is difficult to evaluate the significance of M:F comparisons. Therefore, the authors should be cautious not to over-interpret their data. However, if more clearly presented, the combined differences between the microbiome of ASD and NT participants, are quite compelling and add to the growing body of literature linking various aspects of the ASD phenotype to the gut microbiome. Therefore, if the specific comments below are addressed, this manuscript may be of interest to IJMS readers.
1. The authors should be consistent in how they present ASD statistics. For example, the authors should consider stating the US ASD prevalence at a percentage the population to be consistent with how the Mexico prevalence numbers are presented. The authors should also make sure their statistics are correct and used in the proper context. In addition, the authors state that M:F ASD frequency is 4.5:1 is this in the US, Mexico or the world? According to the data they cite the M:F is estimated from 3.6-4.6:1 depending on the year of assessment, with 4.3:1 being the most recent ratio (Chiarotti 2020), whereas the second study (Bargiela 2017) estimates a 3:1 ratio. The estimated prevalence of autism and the ratio of M:F individuals varies widely between studies. The authors should cites more than one study and use statistics which more accurately represent the range of estimates.
2. In regard to Figure 1: Line 99 states “no significant difference was found” how is significance being calculated? The statistical test used for this analysis should be referenced in the figure legend for quick reference.
3. Data for each of the four genus that are significantly different between ASD and NT individuals should be plotted next to each other as categorical scatter plots with error bars and statistical significance indicated. This approach should be applied to all significant differences shown in the paper to allow the reader to fully evaluate the data. The authors should also provide in-depth discussion and sufficiently cite previous literature that associates these microbes with ASD.
4. The authors state: “In terms of the severity level of core symptoms of autism, 9 individuals (30%) were classified as mild severity, 13 (43%) were classified as moderate severity, and 8 (27%) were classified as severe” Were there significant differences between the microbiomes of ASD patients in each of these severity groups? If so, then this analysis should be added to the paper.
5. The authors state: ‘Of particular note is the exclusive presence of the genus Megamonas in the ASD female group, with a relative abundance of 4.45%. Notably, this genus was absent in ASD males as well as both male and female NT groups.” Megamonas has previously been observed in the guts of non-autistic individuals and therefore the significance of this observation should be discussed in the context of published literature.
6. For clarity, Figure 5 should be annotated with the function of the pathways that are under- vs over-represented, not just the identifier. Further explanation of the findings of figure 5 and their significance should be added to the text.
Author Response
Comments to revisor 2
- The authors should be consistent in how they present ASD statistics. For example, the authors should consider stating the US ASD prevalence at a percentage the population to be consistent with how the Mexico prevalence numbers are presented. The authors should also make sure their statistics are correct and used in the proper context. In addition, the authors state that M:F ASD frequency is 4.5:1 is this in the US, Mexico or the world? According to the data they cite the M:F is estimated from 3.6-4.6:1 depending on the year of assessment, with 4.3:1 being the most recent ratio (Chiarotti 2020), whereas the second study (Bargiela 2017) estimates a 3:1 ratio. The estimated prevalence of autism and the ratio of M:F individuals varies widely between studies. The authors should cites more than one study and use statistics which more accurately represent the range of estimates.
Answer: The references were revised, and the ASD statistics are presented in a more consistent manner. The paragraph was changed to:
"According to a recent report, the prevalence of ASD in 2018 revealed that ASD affects approximately 23 per 1,000 children under 8 years in the United States and 22.5 per 1,000 in Hispanic children [3]. In particular, the prevalence of ASD in Mexico was estimated to be 8.7 cases per 1,000 [4]. ASD is more prevalent in males than females, and in the USA, it is diagnosed more frequently in boys than girls, with the ratio varying by state from 3.3-5.2:1 in 2018 [3]. In England, data from 2016 estimated the male-female ratio to be 3:1 [5], and the most recent ratio obtained from USA data is 4.3:1 [6]."
All the changes were highlighted.
- In regard to Figure 1: Line 99 states “no significant difference was found” how is significance being calculated? The statistical test used for this analysis should be referenced in the figure legend for quick reference.
Answer: we change the sentence with: “no significant difference was found by the Welch’s t-test” in that line and we refer the test in the figure legend.
- Data for each of the four genus that are significantly different between ASD and NT individuals should be plotted next to each other as categorical scatter plots with error bars and statistical significance indicated. This approach should be applied to all significant differences shown in the paper to allow the reader to fully evaluate the data. The authors should also provide in-depth discussion and sufficiently cite previous literature that associates these microbes with ASD.
Answer: We add the suggested plots to all significant differences reported in figures and in the text.
- The authors state: “In terms of the severity level of core symptoms of autism, 9 individuals (30%) were classified as mild severity, 13 (43%) were classified as moderate severity, and 8 (27%) were classified as severe” Were there significant differences between the microbiomes of ASD patients in each of these severity groups? If so, then this analysis should be added to the paper.
Answer: The suggested analysis has been incorporated into the paper. In the "Results" section, a new subsection, 2.4, has been included to elucidate the observed results. Additionally, a new figure (now Figure 3) displaying the graphical results has been added. Furthermore, these results were discussed in the Discussion section. All the changes have been highlighted.
- The authors state: ‘Of particular note is the exclusive presence of the genus Megamonas in the ASD female group, with a relative abundance of 4.45%. Notably, this genus was absent in ASD males as well as both male and female NT groups.” Megamonas has previously been observed in the guts of non-autistic individuals and therefore the significance of this observation should be discussed in the context of published literature.
Answer: The discussion in this section has been revised, and the following sentence has been added:
"Notably, the genus Megamonas was exclusively observed in our ASD female group. However, Megamonas has previously been observed in the guts of non-autistic individuals in other populations (47,48). The absence of Megamonas in the other groups in our study may be attributed to factors such as the small sample size."
The sentences were highlighted.
- For clarity, Figure 5 should be annotated with the function of the pathways that are under- vs over-represented, not just the identifier. Further explanation of the findings of figure 5 and their significance should be added to the text.
Answer: Figure 5 has been updated to Figure 6, and the pathways that are over- and underrepresented are now annotated on the left side next to the identifier. The explanation of these findings is provided in the discussion section.